# Supramolecular Tools to Improve Wound Healing and Antioxidant Properties of Abietic Acid: Biocompatible Microemulsions and Emulgels

**DOI:** 10.3390/molecules27196447

**Published:** 2022-09-30

**Authors:** Alla Mirgorodskaya, Rushana Kushnazarova, Rais Pavlov, Farida Valeeva, Oksana Lenina, Kseniya Bushmeleva, Dmitry Kuryashov, Alexandra Vyshtakalyuk, Gulnara Gaynanova, Konstantin Petrov, Lucia Zakharova

**Affiliations:** 1Arbuzov Institute of Organic and Physical Chemistry, FRC Kazan Scientific Center of RAS, Arbuzov Str. 8, 420088 Kazan, Russia; 2Kazan National Research Technological University, Karl Marx Str. 68, 420015 Kazan, Russia

**Keywords:** drug delivery systems, emulgel, microemulsion, surfactant, gelating polymer, abietic acid, antioxidant activity, wound healing

## Abstract

Abietic acid, a naturally occurring fir resin compound, that exhibits anti-inflammatory and wound-healing properties, was formulated into biocompatible emulgels based on stable microemulsions with the addition of a carbamate-containing surfactant and Carbopol^®^ 940 gel. Various microemulsion and emulgel formulations were tested for antioxidant and wound-healing properties. The chemiluminescence method has shown that all compositions containing abietic acid have a high antioxidant activity. Using Strat-M^®^ skin-modelling membrane, it was found out that emulgels significantly prolong the release of abietic acid. On Wistar rats, it was shown that microemulsions and emulgels containing 0.5% wt. of abietic acid promote the rapid healing of an incised wound and twofold tissue reinforcement compared to the untreated group, as documented by tensiometric wound suture-rupture assay. The high healing-efficiency is associated with a combination of antibacterial activity of the formulation components and the anti-inflammatory action of abietic acid.

## 1. Introduction

Development and improvement of new drugs remains an urgent task of fundamental and practical importance. Successful research in this field requires an investment of broad multidisciplinary expertise and effort in the areas of chemistry, biology and medicine. Considering how much modern drugs have improved in the last few decades, novel medicines need to satisfy the increasingly strict safety, biocompatibility, and targeted activity criteria. In this regard, nanomedicine has been the main driving force which allows the bypassing of drug insolubility, provides protection and efficient transport of active substances in biological conditions, enables penetration of biological barriers, targeted delivery, and lowers adverse effects [1,2,3].

Many of the modern important drugs (atropine, ephedrine, morphine, Taxol, and others) are derived from plants and animals that were already known and have been used in medicine throughout human history. The analysis of active substances in natural compounds can be a fruitful approach to the development of new, efficient medicine [4,5,6]. One of the recent works comprehensively studied in vivo anti-inflammatory and wound-healing properties of pine essential oils and outlined their high efficacy in treating wounded tissue [4]. Although the historic use of pine-derived compounds is documented, only recently scientific attention has focused on abietic acid in particular. It is a natural compound extracted from fir resin, that is commonly used in lacquers, paints, varnishes, and soaps. Abietic acid has possible anticancer [7,8], anti-inflammatory, and wound-healing properties [9,10,11], albeit there is some evidence of its allergenicity [12]. It was shown that abietic acid has anti-inflammatory properties in vivo when administered orally or topically [11]. The in vitro evidence of angiogenesis’ stimulation as well as in vivo wound-healing properties of abietic acid are documented in [10].

However, little to no research involving complex abietic acid formulations for topical application such as creams, ointment, lotions, or gels has been conducted. Forms that contain large amounts of oily bases (petrolatum, beeswax, vegetable oils) show a delayed release of hydrophobic substrates and have a thick and greasy texture. The use of the gel form avoids these disadvantages [13]. Common gelating agents used for this purpose are often represented by weakly cross-linked synthetic and natural polymers, containing polyalcohol and polysaccharide residues.

When applied transdermally, the combination of emulsion and gel leads to a number of improvements in key indicators: (1) increased stability, (2) accelerated release of hydrophobic substances and their penetration through the skin, (3) improved drug loading efficiency, and (4) increased retention time for emulsions on the skin surface [14,15,16,17]. Emulgels are mostly loaded with non-steroidal anti-inflammatory drugs, for example, valdecoxib [18], nimesulide [19], meloxicam [20], ketoprofen [21], and diclofenac sodium [22]. In addition, the results of studies on emulgels containing the flavonoid quercetin [16,23], as well as extracts isolated from plants [24,25], are described. As a basis for the formation of emulgels, microemulsions are widely used, which usually consist of an aqueous and oily (hydrocarbon) phase separated by a layer of micelle-forming surfactants, sometimes including co-surfactants and which are a special case of emulsions with a droplet size of less than 100 nm [26,27]. Microemulsions are macroscopically homogeneous, thermodynamically stable, dispersed systems exhibiting a high solubilization effect with respect to both lipophilic and hydrophilic compounds [26,28,29], which are not disturbed when gelling agents are added to them.

In the formation of microemulsions and similar systems intended for biomedical applications, nonionic surfactants are often used, which have low toxicity and are compatible with most biologically active substances [30,31]. However, in terms of their effectiveness, they are inferior to more toxic ionic surfactants. The combination of a nonionic surfactant with a cationic surfactant can impart a positive charge to systems, which improves their stability, enhances the solubilization effect, and also facilitates the interaction of the formed systems with negatively charged areas of a living cell. The presence of cationic surfactants in nanosized medicinal compositions gives them additional efficiency, especially when used as ophthalmic and transdermal preparations [32,33,34].

In the present work, the task was set to use microemulsions and emulgels as systems that would reveal the potential of abietic acid as a wound healing and antioxidant agent. In the course of the work, first, biocompatible microemulsions based on oleic acid and Tween 80 [35,36,37] were formed. To obtain stable emulgels containing a high concentration of abietic acid, Carbopol^®^ 940, a gelating cross-linked polymer was added to the microemulsions. When optimizing the composition of microemulsions, not only the ratio of components was varied, but it was also modified with the addition of a cationic surfactant containing a carbamate fragment in the head group—N-[2-((butylcarbamoyl)oxy)ethyl]-N,N-dimethylhexadecaneammonium bromide (CB-16(Bu)). The appeal to cationic surfactants of this class is caused primarily by the fact that carbamate-containing surfactants can overcome various biological barriers, and this may be a favorable factor in the creation of delivery systems.

The structural formulas of the compounds in the study are presented below (Figure 1).

## 2. Results and Discussion

### 2.1. Dimensional Characteristics and Viscosity Properties of Microemulsions and Emulgels

When choosing the basic microemulsion (ME 1), biocompatible compositions described earlier in the literature [35] were considered with slightly changed ratios of ingredients. The aqueous phase of the microemulsion, represented by phosphate-buffered saline (PBS), is separated from the oil phase (oleic acid) by a non-ionic surfactant, Tween 80. Ethanol is distributed between the aqueous phase and the surface layer of microemulsion droplets and acts as a co-solvent and a co-surfactant. To impart a positive charge to the nanoparticles, a part of Tween 80 was replaced by a cationic carbamate surfactant, which is significantly less toxic than common cationic surfactants: LD50 for CB-16(Bu) it is 82 mg/kg, while for cetyltrimethylammonium bromide its value is 27 mg/kg (mice, intraperitoneal administration) [38]. It is important that the compounds carrying a carbamate fragment have the ability to overcome biological barriers, which was reflected in a number of publications. For example, the authors of [39] reported the synthesis and biological evaluation of molecular transporters, oligocarbamates, which ensured efficient absorption by cells and improved the permeability of biotin (model probe) through the stratum corneum of the skin. The studies on the creation of cationic lipids containing a carbamate fragment that act as non-viral vectors for the delivery of gene material are also noteworthy. Liposomes with low toxicity based on these lipids demonstrated high efficiency of transfection into HeLa (Human cervical cancer), HepG-2 (Human hepatocellular carcinoma), and NCI-H460 (Human lung cancer) cells [40,41]. Their ability to penetrate cell membranes and to transport solubilized substances was detected using fluorescence microscopy on Chang liver cells [42], as well as in experiments in vivo when testing the anti-inflammatory effects of the drug indomethacin loaded into nano- and microemulsions, which included carbamate surfactants [36]. The compositions of the systems in study and corresponding abbreviations are listed in Table 1.

As a result of comparative experiments, in which the size and degree of polydispersity of particles in the system, its stability and solubilization capacity with respect to abietic acid, as well as the ease of its conversion into emulgels were evaluated, two main microemulsions (ME 1 and ME 3) were identified, the composition of which are given in Table 1. It follows from the DLS data that the introduction of a cationic surfactant into the initial microemulsion leads to some decrease in the particle size, but the addition of abietic acid practically does not change their hydrodynamic diameter (Table 2, Appendix A). The ζ-potential of the microemulsion droplets without carbamate surfactant was about 0 mV, and it increased to 6–9 mV in its presence (Appendix A). The addition of abietic acid does not affect the value of this parameter.

Maximal solubility of abietic acid in the ME 1 and ME 3 systems was estimated by spectrophotometry from the intensity of the absorption band at 240 nm (molar extinction coefficient 13,000 l⋅mol^−1^⋅cm^−1^). The choice of the analytical signal was carried out, taking into account the spectra of abietic acid recorded in various media under conditions of varying its concentration (Appendix A, Appendix A). It was found that approximately up to 1.2% wt. of abietic acid is soluble in ME 1, and with a further increase in its content in the system, phase separation occurs. In a microemulsion containing a carbamate surfactant (ME 3), up to 1.8% wt. of abietic acid can be dissolved without decomposition (delamination) of the system, while some increase in its viscosity is observed. In further studies, a constant concentration of abietic acid was maintained, 0.5% wt. (ME 2, ME 4) and 1% wt. (ME 5). The resulting microemulsions are stable and do not change their characteristics for at least a month (Table 2).

Viscosity is an important parameter for a topical drug delivery system. It affects drug release and absorption, and thus affects the therapeutic benefit from the formulation [43]. The rate of drug release decreases with an increase in the concentration of the gelling agent because the viscosity of the formulation increases [44]. In addition, increased viscosity allows the formulation to stay on the skin surface for longer, which also has a positive effect on its effectiveness. In this work, we used Carbopol^®^ 940 as a thickener, and with its help, on the basis of the above microemulsions, we obtained a number of homogeneous viscous emulgels (EG 1—EG 4), which retained their properties for more than a month. Carbopol^®^ 940 is a commercially available cross-linked polyacrylate polymer that acts as a high performance and low-dose thickener. It is a hydrophilic, non-toxic, and non-irritating synthetic polymer often used in the pharmaceutical industry as a base for soft dosage forms. The advantage of this polymer is its ability to form gels in water at room temperature within a wide pH range (from 4 to 10) [45]. The amount of Carbopol^®^ 940 affects the consistency of the composition [14]. When choosing the concentration of Carbopol^®^ 940 in emulgels, we primarily focused on the literature data. In a significant part of the published works, Carbopol^®^ 940 is used as a thickener at a concentration of 0.5% wt. [16,46,47].

The change in rheological properties when adding Carbopol^®^ 940 to the microemulsion can be demonstrated on the example of EG 4 and ME 4 (Figure 1 and Appendix A). In the absence of a polymer, the viscosity of the microemulsion does not exceed 0.1 Pa·s and weakly depends on the shear rate, i.e., the behavior is Newtonian. However, when Carbopol^®^ 940 is added, the viscosity of the microemulsion increases by several orders of magnitude, and the nature of the rheological behavior changes to pseudoplastic: at low shear rates, the viscosity is very high and changes insignificantly, but then, with increasing shear rate, it drops sharply (Appendix A).

As can be observed, different responses were obtained for the systems under study. In the case of the microemulsion (ME 4), the loss modulus (G″) is significantly higher than the storage modulus (G′), and the formulations do not really result in gels but in liquid-like dispersions. A tendency to reach a crossover point between these moduli was observed at high frequencies (Figure 1a). This corresponds to the dynamic characteristics of a fluid without entanglements. On the other hand, for EG 4, G′ becomes higher than G″ and is practically unchanged in the whole frequency range. This indicates the formation of elastic gels. As shown by the authors of [48], the gel strength of biopolymer-dispersed systems, from dilute solutions to crosslinked gels, can be quantified from small amplitude oscillatory shear measurements as a function of the G′ and G″ frequency dependence. In this case, the characteristic parameters are the slopes of G′ and G″ versus the frequency plots, and the relative values of both viscoelastic functions, i.e., the relative elasticity, expressed in terms of the loss tangent (tan δ = G″/G′). Regarding the evolution of the loss tangent with frequency (Figure 1b), the EG 4 shows the lower values of the loss tangent indicating a higher relative elasticity due to the high-level physical entanglements. On the other hand, ME 4 displays the highest values of the loss tangent, higher than 1 in the whole frequency range indicating an essentially viscous behavior characteristic of polymer fluids without a significant formation of physical entanglements [49].

### 2.2. Prerequisites for the Use of Microemulsions and Emulgels Loaded with Abietic Acid in Biomedical Applications

#### 2.2.1. Sustained Release of Abietic Acid

The process of abietic acid release from microemulsions and emulgels was monitored using dialysis followed by spectrophotometric determination of the optical density at 240 nm. In Figure 2, the release profiles of free and formulated abietic acid are displayed, which indicate that each of the studied formulations significantly lowers the rate of release. After 24 h, more than 80% of free abietic acid leaves the dialysis bag, while the release from microemulsions and emulgels tops off at 11–15%. Significant differences between microemulsions and emulgels are not present; however, for compositions containing CB-16(Bu), the release is slightly lower.

The obtained kinetic profiles (Figure 2) were approximated using different mathematical models, traditionally used to evaluate release profiles: zero and first order, Higuchi, and Korsmeyer–Peppas [50,51,52]. The results are shown in Table 3. Along with the correspondence of approximated curves to data points, the main criteria used to evaluate model applicability was the correlation coefficient (*R^2^*). The obtained experimental results are best described by the Korsmeyer–Peppas equation: *R*^2^ ≥ 0.99 (Table 3). The value of *n* derived from the equation parameters allows the classification of the release mechanics [53]: free abietic acid corresponds to *n* a little over 0.5, which indicates Fickian diffusion. For microemulsions and emulgels, the value of *n* is near 1, which implies essentially zero-order release, or, in polymer science terms, case-II transport, for at least the first 8 h of application [52]. The release from microemulsions and emulgels is prolonged, and is also constant over time, which is a sought-after characteristic for drug delivery systems, that allows the better control of the absorbed drug concentration.

Since the obtained microemulsions and emulgels loaded with abietic acid were designed as dermal delivery systems, their permeating action was studied using Franz cells and Strat-M^®^ model membranes imitating human skin. It was shown that, in the case of emulgels (EG 2 and EG 4), the passage of abietic acid through the membrane is slower than for microemulsions (ME 2 and ME 4) (Figure 3). The addition of a carbamate surfactant to the composition (ME 4 and EG 4) further prolongs the passage of the active substance through the model membranes. Probably, an electrostatic attraction occurs between the cationic surfactant and abietic acid, which is mostly present in anionic form in buffer conditions, which impedes the release. This property of the prepared formulations for surface application makes it possible to provide a long-term uniform effect of the active substance on the affected area after only a single application.

To evaluate the effect of studied compositions on the permeability of the drug, the apparent permeability coefficient (*P_app_*) was calculated for the abietic acid formulated in ME and EG. *P_app_* was obtained from the slopes of the curves presented in Figure 4, in accordance with Equation (1). *P_app_* was higher for abietic acid from microemulsions that from emulgels, which is likely tied to the viscosities of formulations (Figure 4). An increase in the viscosity leads to a decrease in the drug permeation through the skin, which was also observed, for example, in microemulsions and gels loaded with curcumin and repaglinide [54,55].

#### 2.2.2. Antioxidant Properties of Abietic Acid Containing Compositions

Samples of microemulsions and emulgels were tested for antioxidant activity using a commonly used chemiluminescent model system, including an aqueous solution of luminol and 2,2′-azobis(2-methylpropionamidine) dihydrochloride [56]. Figure 5 shows the CL kinetics in the systems under study. The maximum chemiluminescence intensity (CL amplitude) and the time from the moment of administration of AAPH to the onset of luminescence development (latent period) were chosen as the measured parameters. The period of CL induction in the presence of an antioxidant can be considered as the time required for its inactivation in the process of interaction with the radical initiators formed in the systems. Obtained data show that compositions without abietic acid (ME 1, ME 3, EG 1 and EG 3) insignificantly inhibit oxidizing agents, while the introduction of abietic acid (ME 4, ME 5, EG 4) imparts pronounced antioxidant properties to the systems. The ability to scavenge free radicals in emulgels is somewhat higher than in microemulsions, and the presence of CB-16(Bu) in the systems slightly increases the antioxidant effect. Obviously, antioxidant properties increase as the abietic acid concentration increases (Figure 5a,b). Among the tested samples, the ME 5 microemulsion containing 1% wt. of abietic acid showed the highest antioxidant activity. Only 10 µL of abietic acid led to a decrease in the intensity of chemiluminescence to almost 0, and this low level of chemiluminescence persisted for 2000 s (Figure 5b).

#### 2.2.3. Wound-Healing Activity of Microemulsions and Emulgels

It was suggested that the obtained microemulsions and emulgels may have a wound-healing effect. It was based on the following circumstances: (1) compositions with antioxidants have the ability to protect living organisms from oxidative stress, while providing anti-inflammatory, antiallergic, antidiabetic, and wound healing effects [57]; (2) oleic acid and alcohol have an antiseptic and antimicrobial effects, which prevent the growth of pathogenic microorganisms in the wound and eliminate their negative effect on tissue regeneration [58,59]; (3) carbamate surfactants have the ability to cross biological membranes and facilitate dermal drug delivery [36]. Such a set of properties promises to provide a positive result in wound therapy. In this regard, the wound-healing effect of microemulsions and emulgels modified with the addition of carbamate surfactants, including those containing 0.5% wt. of abietic acid, was tested using the measurement of skin tensile strength in an incision wound model. Tensile strength is the resistance to breaking under tension. It indicates how much the repaired tissue resists breakage under tension and the rate of skin reparation. For the quantitation, one of the ends of a piece of rat skin that includes a scar was fixed while applying a measurable force to the other one. An increase in the weight required to rupture the tissue indicates an increase in the rate of wound healing. The weight in grams required to disrupt the wound was measured on day 11 post wounding in the control group of rats and after 10 days of treatment with the studied compositions.

The results are presented in Figure 6. It has been established that all of the studied compositions contribute to wound healing: the tensile strength of the tissue is approximately doubled compared to the control. It can be assumed that the viscoelastic properties of emulgels facilitate the application of the therapeutic composition and allow it to be better retained on the wound surface.

The results obtained allow us to state that the microemulsions and emulgels containing abietic acid, obtained based on biocompatible components and modified with CB-16(Bu), are very effective in the treatment of the incised wound.

## 3. Materials and Methods

### 3.1. Materials

Polyethylene glycol sorbitan monooleate (Tween 80, Sigma Aldrich, Saint Louis, MO, USA, ≥99%), oleic acid (Alfa Aesar, Ward Hill, MA, USA, 99%), Carbopol^®^ 940 (Acros Organics, Geel, Belgium), abietic acid (Sigma Aldrich, Saint Louis, MO, USA, ~75%), 2,2′-azobis(2-methylpropionamidine) dihydrochloride (AAPH) and luminol were obtained from Stanchem Sp. z o.o. Przedsiębiorstwo Chemiczne. Reagents were used as received without further purification. The carbamate surfactant CB-16(Bu) was synthesized according to the method described in [38].

### 3.2. Preparation of Microemulsions and Emulgels

Microemulsions were produced by sequentially mixing oleic acid (15.7% wt.), surfactants (19.7% wt.), ethanol (27.2% wt.), and phosphate buffer saline (PBS, 50 mM, pH 6.86) (37.4% wt.). As a surfactant, either only Tween 80 or a mixed composition was used, in which 3% wt. of a nonionic surfactant was replaced by the cationic carbamate surfactant CB-16 (Bu). The resulting microemulsions were used to obtain systems containing 1 or 0.5% wt. of abietic acid. Emulgels were obtained as described in [21] by mixing a microemulsion with a gel formed by 1% wt. solution of Carbopol^®^ 940 in PBS and vigorous vortexing until a homogenous thick liquid was formed. The fabricated microemulsions and emulgels were stored at 25 °C.

### 3.3. Particle Size and ζ-Potential Determination

The hydrodynamic diameter and ζ-potential of the drops in microemulsions were analyzed by dynamic light scattering technique. Zetasizer Nano ZS (Malvern Instruments, Malvern, UK) was used at a scattering angle of 173° using a He-Ne laser (λ = 633 nm). Measurements were carried out at 25 °C. The pulse accumulation time was 5–8 min. Each measurement was an average of 16 runs. The signals were analyzed using a single-plate multichannel correlator coupled with a computer equipped with the software package for the evaluation of effective hydrodynamic diameter of dispersed particles. All samples were analyzed in triplicate; the average error of measurements was approximately 4%.

### 3.4. Rheological Studies of Microemulsions and Emulgels

Haake RheoStress 6000 rheometer (Germany) was used for the rheologic characterization. For solutions with low viscosity a coaxial cylinders’ measuring cell with a double gap was used (21.7 mm outer cylinder, 18 mm inner cylinder, 55 mm height). Measurements of viscous samples were conducted using a 35 mm cone-plane measuring configuration with a cone angle of 2° and a gap maintained at 0.105 mm. Two types of experiment were performed. First, frequency sweep (dynamic experiment) was performed and the storage, G′, and loss, G″, moduli were measured, as functions of the frequency of oscillations, varied between 0.001 and 10 Hz. These measurements are in the linear viscoelastic regime, as determined previously by dynamic strain–sweep measurements. Second, in the static mode (constant shear) the experiments were carried out in the stress range from 0.002 to 100 Pa. The viscosity of the solutions *η* was determined as the relation of shear stress to shear rate $\eta =\sigma /\dot{\gamma }$. In the range of low rates, the viscosity reached a plateau (it did not depend on stress), which was taken as the value of Newtonian viscosity at zero shear rate *η_0_*.

### 3.5. Determination of Abietic Acid Solubility

The UV spectra of abietic acid were recorded using the Specord 250 Plus (Analytik Jena, Jena, Germany) at 200–600 nm. The limiting solubility of abietic acid was determined spectrophotometrically at 240 nm by analogy with [38]. Abietic acid concentration in the solution was calculated by the Lambert–Beer equation. The measurements were carried out using reference cuvettes containing the systems under study without additions of abietic acid. The experiments were performed in triplicate.

### 3.6. Dialysis

To evaluate the cumulative release of abietic acid, microemulsion or emulgel (3 g) containing abietic acid (0.5% wt.) were placed in a 3.5 kDa dialysis bag. The bag was immersed in the solution containing 25 mL of phosphate buffer (pH 6.86) and ethanol (25 mL) at temperature 34 °C. At certain intervals, samples were taken from the external solution. The abietic acid concentration was measured using a Specord 250 Plus spectrophotometer (Analytik Jena, Germany) as described in Section 3.5. Obtained kinetic data were analyzed using the following mathematical models: zero (*Q* = *k*_0_*t*) and first order (ln(*1-Q*) = −*k*_1_*t*), Higuchi (*Q* = *k*_H_*t*^1/2^), and Korsmeyer–Peppas (*Q* = *k*_KP_*t*^n^), where *Q* is the fraction of drug released at time *t*; k—is the release constant; *n*—is diffusion release exponent [53].

### 3.7. Abietic Acid Diffusion and Permeation Studies

Vertical Franz diffusion cells (SES GmbH Analysesysteme, Leonberg, Germany) were used to model and compare abietic acid release and diffusion from microemulsions (emulgels). On the Franz cells, the donor and acceptor compartment were separated by the synthetic membrane, Strat-M (Merck Millipore, Burlington, MA, USA). A total of 0.25 g of prepared formulation was placed in the donor chamber. PBS:ethanol (1:1) solution was used as the acceptor phase. The receptor compartment with a volume of 5 mL was maintained at a constant temperature (34 °C) and stirring. At certain time intervals, the concentration of the abietic acid was measured spectrophotometrically with Specord 250 Plus (Analytik Jena, Jena, Germany) spectrometer. The quantity of the permeated abietic acid was expressed as mg/cm^2^ units after three repetitions of the experiment.

Apparent permeability coefficient (*P*_app_) was calculated as the slope of linear part of permeability curve according to Equation (1):(1)$$P_{\mathrm{app}}=\frac{dQ}{dt}\frac{V}{A\cdot C_{0}}$$
where *dQ/dt* is the speed of abietic acid concentration change in the acceptor compartment, mg/s; *A*—Area of the membrane, cm^2^; *V*—Volume of accepting compartment (Franz cell), cm^3^; *C_0_*—Initial mass of loaded abietic acid, mg.

### 3.8. Antioxidant Activity

Antioxidant properties were evaluated by the ability of microemulsions and emulsifier to scavenge free radicals formed in the reaction mixture, containing 2,2′-azobis(2-methylpropionamidine) dihydrochloride and luminol. The mechanism of this system is that in which an aqueous medium AAPH undergoes thermally induced degradation with the formation of peroxyl radicals, which can interact with luminol which in turn acts as a radical indicator, while chemiluminescence (CL) is recorded. The CL was monitored [56] with a Lum-100 chemiluminometer (DISoft, Zarechny, Russia). To the cuvette of the chemiluminometer, thermostat set at 30 °C, containing 1 mL of the reaction mixture of AAPH, luminol, and 0.5 M Tris buffer (pH 8.6), 5 or 10 μL of test emulsions were added. After addition of substances, the intensity of CL was recorded for 6000 s. Results were expressed in % relative to the baseline. The experiments were repeated three times.

### 3.9. Incision Wound Model

All experiments with animals were carried out in accordance with the Directive of the Council of the European Union 2010/63/EU. The protocol of experiments was approved by the Animal Care and Use Committee of FRC Kazan Scientific Center of RAS. Experiments were performed using Wistar rats (weighing 250–300 g, 12-week old) of both sexes. Animals were purchased from the Laboratory Animal Breeding Facility (Branch of Shemyakin-Ovchinnikov Institute of Bioorganic Chemistry, Puschino, Russia). The animals were kept in sawdust-lined plastic cages at 20–22 °C in a 12 h light/dark cycle, 60–70% relative humidity, and given ad libitum access to water and food.

The animals were deeply anesthetized with isoflurane, the dorsal skin hairs of animal were removed with an electrical clipper and 3 cm long paravertebral incisions were made through the full thickness of the skin on each side of the vertebral column [60]. After incision, the parted skin was surgically sutured 0.5 cm apart using a surgical thread and curved needle. The wound was left undressed. The skin around the wound was treated with chlorhexidine. Rats were randomized into five groups consisting of six rats per group. The microemulsions and emulgels were applied using a Pasteur pipette, daily until 10 days (4 drops 2 times a day, morning and evening). The sutures were removed on the day 11. The rats were euthanized and the tensile strength of cured wound skin was measured using a tensiometer as described [61].

All results were presented as means ±SD for six animals per group. Statistical significance was evaluated by ANOVA followed by post hoc test at *p* < 0.05.

## 4. Conclusions

Thus, based on microemulsion compositions modified with carbamate surfactants, emulgels containing the active compound of fir resin, abietic acid, were obtained. It has been established that the complex viscosity modulus of the emulgel is several orders of magnitude higher than the viscosity of microemulsions, which indicates the formation of a three-dimensional spatial network in the solution. Franz cell diffusion showed that the release of abietic acid is slowed down in microemulsions and, to an even greater extent, in emulgels, especially in systems containing additives of carbamate surfactants. This makes it possible to ensure a uniform and prolonged intake of the active substance when using the proposed compositions in therapy. Composition-optimized samples were tested for external use as antioxidant and wound-healing compositions. The presence of abietic acid imparts pronounced antioxidant properties to the systems, which is important for rapid wound healing. Therefore, compositions with abietic acid were tested on a rat model of an incision wound, which demonstrated that microemulsions and emulgels containing 0.5% wt. of abietic acid promoted the healing of an incised wound on the skin of rats, almost doubling the force required to break the wound suture. Thus, obtained biocompatible microemulsions and emulgels loaded with abietic acid showed high efficiency when used as antioxidant and wound-healing systems, which opens up broad prospects for their use in medical and biotechnological applications.

## Figures and Tables

**Scheme 1 molecules-27-06447-sch001:**
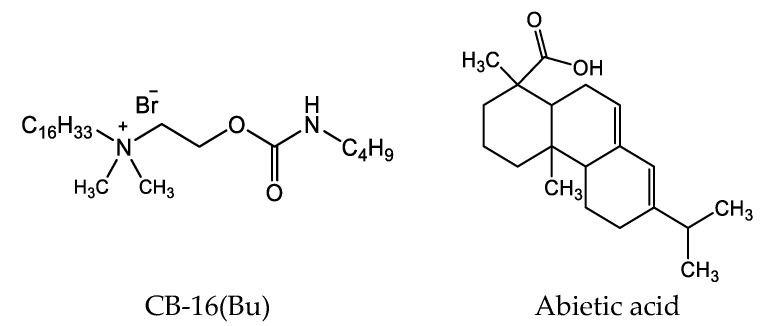
Structural formulas of compounds in study.

**Figure 1 molecules-27-06447-f001:**
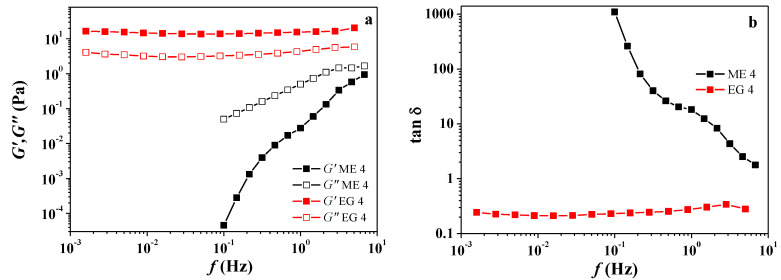
Frequency dependence of the storage (G′) and loss (G″) moduli (**a**); the loss tangent for ME 4 and EG 4 (**b**).

**Figure 2 molecules-27-06447-f002:**
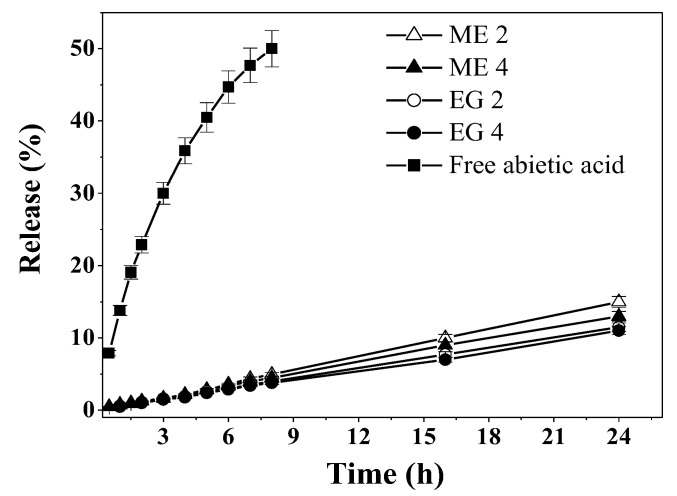
Release of free abietic acid and abietic acid formulated in microemulsions and emulgels. The drug concentration in samples is 0.5% wt., PBS:ethanol (1:1) medium, 34 °C.

**Figure 3 molecules-27-06447-f003:**
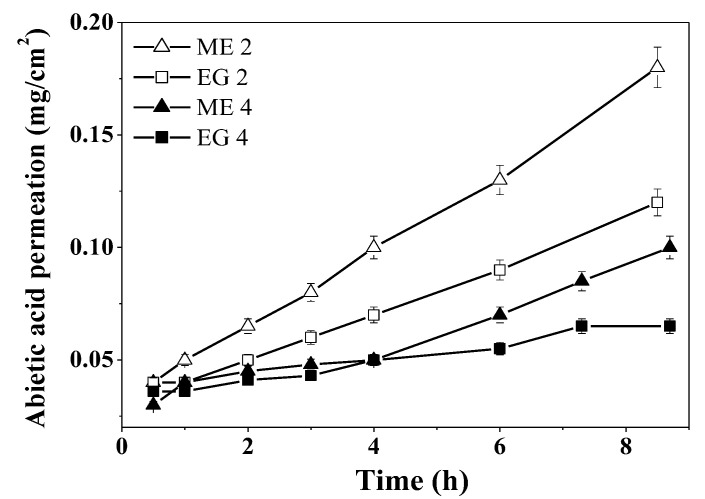
Permeation of abietic acid from the microemulsions and emulgels through Strat-M^®^ membranes over time, PBS:ethanol (1:1) medium, 34 °C.

**Figure 4 molecules-27-06447-f004:**
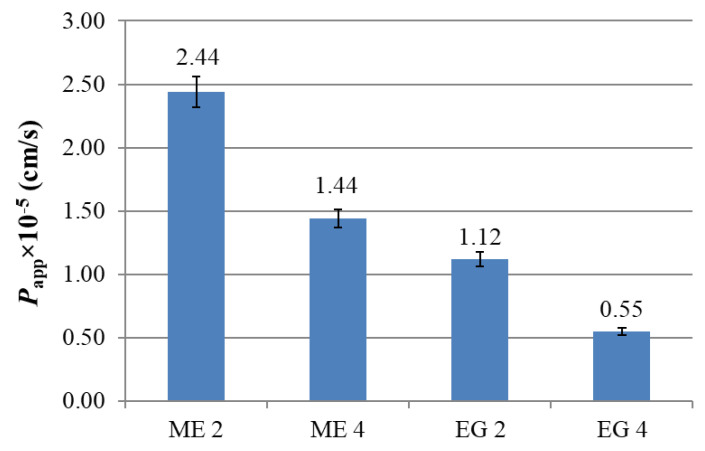
Apparent permeability constants of abietic acid loaded in microemulsions or emulgels.

**Figure 5 molecules-27-06447-f005:**
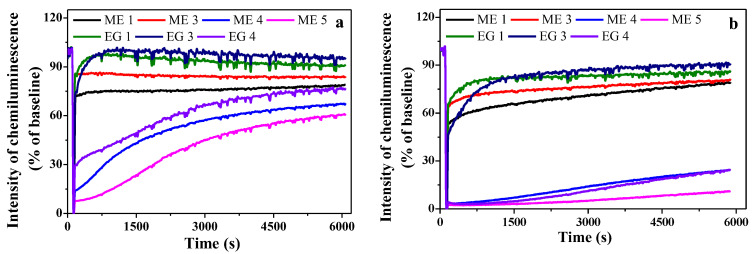
Graph of the change in the level of chemiluminescence in time with the addition of 5 µL (**a**) and 10 µL (**b**) of the tested microemulsions or emulgels in 1 mL of luminol solution.

**Figure 6 molecules-27-06447-f006:**
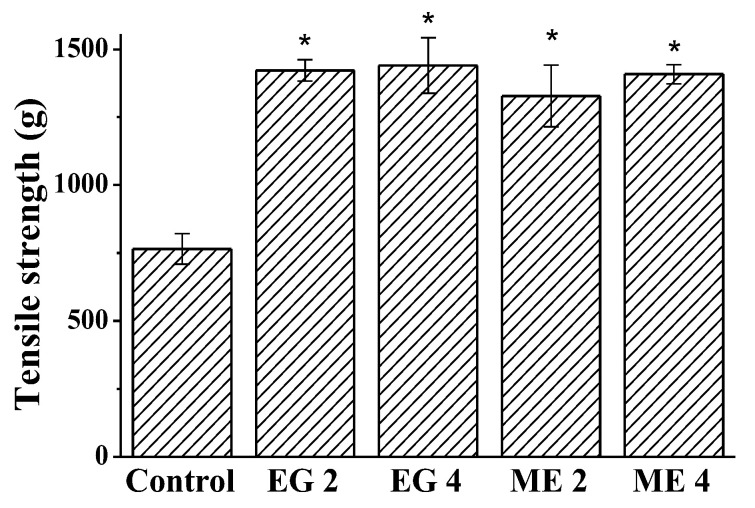
Tensile strength of healing wounds in control group and after treatment with different compositions, * *p* < 0.05 compared to control.

**Table 1 molecules-27-06447-t001:** Composition and designations of the formed microemulsions and emulgels.

System	Concentration, % wt.						
	Oleic Acid	Tween 80	CB-16(Bu)	PBS	Ethanol	Abietic Acid	Carbopol^®^ 940
ME 1	15.7	19.7	-	37.4	27.2	-	-
ME 2	15.6	19.6	-	37.2	27.1	0.5	-
ME 3	15.7	16.7	3.0	37.4	27.2	-	-
ME 4	15.6	16.6	3.0	37.2	27.1	0.5	-
ME 5	15.5	16.5	3.0	37.0	27.0	1.0	-
EG 1	7.8	9.8	-	68.4	13.5	-	0.5
EG 2	7.7	9.7	-	68.2	13.4	0.5	0.5
EG 3	7.8	8.3	1.5	68.4	13.5	-	0.5
EG 4	7.7	8.2	1.5	68.2	13.4	0.5	0.5

**Table 2 molecules-27-06447-t002:** Stability of the microemulsions over time: the size and degree of polydispersity (PdI) of particles (25 °C).

Microemulsion	First Day		A Month Later	
	Diameter, nm	PdI	Diameter, nm	PdI
ME 1	122 ± 6	0.127 ± 0.017	110 ± 5	0.145 ± 0.018
ME 2	118 ± 5	0.179 ± 0.021	115 ± 5	0.188 ± 0.019
ME 3	60 ± 3	0.285 ± 0.031	70 ± 3	0.265 ± 0.033
ME 4	62 ± 3	0.276 ± 0.028	72 ± 4	0.282 ± 0.030

**Table 3 molecules-27-06447-t003:** Curve fitting results (Figure 2) for the different release models.

Formulation	Zero Order		First Order		Korsmeyer–Peppas			Higuchi	
	*R^2^*	*k*	*R^2^*	*k*	*R^2^*	*k*	*n*	*R^2^*	*k*
Free abietic acid	0.7969	7.41 ± 0.46	0.9323	0.10 ± 0.005	0.9912	15.0 ± 0.62	0.60 ± 0.02	0.9742	17.5 ± 0.39
ME 2	0.9984	0.62 ± 0.005	0.9962	0.0065 ± 9.3·10^−5^	0.9988	0.61 ± 0.02	1.03 ± 0.01	0.7621	2.12 ± 0.24
ME 4	0.9983	0.55 ± 0.005	0.990	0.0058 ± 4.0·10^−5^	0.9989	0.57 ± 0.02	0.97 ± 0.01	0.7912	1.91 ± 0.19
EG 2	0.9986	0.48 ± 0.004	0.9994	0.0050 ± 2.7·10^−5^	0.9999	0.53 ± 0.001	0.97 ± 0.001	0.7829	1.70 ± 0.18
EG 4	0.9976	0.45 ± 0.005	0.9979	0.0047 ± 5.0·10^−5^	0.9980	0.50 ± 0.02	0.97 ± 0.02	0.7792	1.61 ± 0.17

## Data Availability

The data presented in this study are available on request from the corresponding author: Rushana Kushnazarova.

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
