# Peer review of "Supramolecular Tools to Improve Wound Healing and Antioxidant Properties of Abietic Acid: Biocompatible Microemulsions and Emulgels"

_molecules, 2022, doi:10.3390/molecules27196447_

Round 1
Reviewer 1 Report
Development and improvement of new drugs remains an urgent task of fundamental and practical importance. Successful research in this field requires an investment of broad multidisciplinary expertise and effort in the areas of chemistry, biology and medicine. In the work, biocompatible microemulsions and emulgels have been used as systems that would reveal the potential of abietic acid as a wound healing and antioxidant agent. This is within the scope of the journal Molecules and answers the criteria of scientific interest and novelty. From my point of view, the work is well-done, results obtained are reliable and useful. Therefore, I recommend the manuscript for the publication after minor revision based on the following comments:
1. Introduction: “The appeal to cationic surfactants of this class is caused primarily by the fact that carbamate-containing surfactants can overcome various biological barriers, and this may be a favorable factor in the creation of delivery systems.” Please explain how exactly the carbamate fragment is able to facilitate penetration through biological barriers? Please, support your answer with references.
2. It should be explained why Carbopol® 940 was chosen among a significant number of gelling polymers? Did the carbapol content vary during the formation of emulgels based on microemulsions? Stability of microemulsions is discussed in the manuscript, however, such data for emulgels are not given? It should be discussed, whether the resulting emulgels retain their gel-like state during long-term storage.
3. Few DLS data are available in the manuscript. Meanwhile they are important not only for characterizing the particle sizes, but also for monitoring their stability. I recommend to transfer the figure or table with dimensional characteristics from SI section to the main part of the manuscript, and give detailed information on this method in the experimental part.
4. For convenience, I recommend to move the table, which shows the compositions of the microemulsions and emulgels, as well as their designations, from the experimental part to the discussion of the results.
Minor:
1. “As a result of comparative experiments, in which the size and degree of polydispersity of particles in the system, its stability and solubilization capacity with respect to abietic acid, as well as the ease of its conversion into emulgels, were evaluated, two main microemulsions (ME 1 and ME 3), composition for which are given in Table 1.”
It should be table 2, please revise.
2. It follows from the DLS data that the introduction of a cationic surfactant into the initial microemulsion leads to some decrease in the particle size, but the addition of abietic acid practically does not change their hydrodynamic diameter (Table S1, Figure S1,S2).
It should be Figures S1,S2.
3. In the table S1 extinction coefficient is equal to12700, but in the text the value of 13000 is given? Bring these in line.
4. Did the authors measure the cumulative release?Add details to Materials and Methods if applicable.
Author Response
Response of authors to reviewer comments
Manuscript ID: molecules-1918422
Title: Supramolecular tools to improve wound healing and antioxidant properties of abietic acid: biocompatible microemulsions and emulgels
Journal: Molecules
Corresponding Author: Dr. Kushnazarova
Authors: Alla Mirgorodskaya, Rushana Kushnazarova*, Rais Pavlov, Farida Valeeva, Oxana Lenina, Ksenia Bushmeleva, Dmitry Kuryashov, Alexandra Vyshtakalyuk, Gulnara Gaynanova, Konstantin Petrov, Lucia Zakharova
Reviewer 1
Development and improvement of new drugs remains an urgent task of fundamental and practical importance. Successful research in this field requires an investment of broad multidisciplinary expertise and effort in the areas of chemistry, biology and medicine. In the work, biocompatible microemulsions and emulgels have been used as systems that would reveal the potential of abietic acid as a wound healing and antioxidant agent. This is within the scope of the journal Molecules and answers the criteria of scientific interest and novelty. From my point of view, the work is well-done, results obtained are reliable and useful. Therefore, I recommend the manuscript for the publication after minor revision based on the following comments:
- Introduction: “The appeal to cationic surfactants of this class is caused primarily by the fact that carbamate-containing surfactants can overcome various biological barriers, and this may be a favorable factor in the creation of delivery systems.” Please explain how exactly the carbamate fragment is able to facilitate penetration through biological barriers? Please, supportyouranswerwithreferences.
Author response:
In the literature, the mechanisms of penetration of compounds carrying a carbamate fragment through various biological barriers are practically not discussed. However, their ability to overcome biological barriers is supported by a number of publications. For example, the work [1] reports synthesis and biological evaluation of molecular transporters, oligocarbamates, which ensure efficient absorption by cells and improve the permeability of biotin (model probe) through the stratum corneum of the skin. Noteworthy are studies on the creation of cationic lipids containing a carbamate fragment that act as non-viral vectors for the delivery of gene material. Liposomes with low toxicity based on these lipids demonstrated high efficiency of transfection into HeLa (Human cervical cancer), HepG-2 (Human hepatocellular carcinoma), NCI-H460 (Human lung cancer) cells [2,3]. In the recent works of our research team, considerable attention has been paid to the synthesis and study of cationic carbamate surfactants. Their ability to penetrate cell membranes and to transport solubilized substances was detected using fluorescence microscopy on Chang liver cells [4], as well as in experiments in vivo when testing the anti-inflammatory effect of the drug indomethacin loaded into nano- and microemulsions, which include carbamate surfactants [5]. Note that the carbamate group is a part of many drugs regulating the activity of cholinesterase for the treatment of CNS diseases, which must be able to pass the blood-brain barrier to produce therapeutic activity [3,6,7].
This information is given in revised manuscript, page 3.
- ItshouldbeexplainedwhyCarbopol® 940 waschosenamongasignificantnumberofgellingpolymers? Didthecarbapolcontentvaryduringtheformationofemulgelsbasedonmicroemulsions? Stabilityofmicroemulsionsisdiscussedinthemanuscript, however, suchdataforemulgelsarenotgiven? Itshouldbediscussed, whethertheresultingemulgelsretaintheirgel-likestateduringlong-termstorage.
Author response:
Carbopol® 940 is a commercially available cross-linked polyacrylate polymer that acts as a high performance and low dose thickener. It is a hydrophilic, non-toxic, non-irritating synthetic polymer often used in the pharmaceutical industry as a base for soft dosage forms. The advantage of this polymer is its ability to form gels in water at room temperature within a wide pH range (from 4 to 10) [8]. The amount of Carbopol® 940 affects consistency of the composition [9]. When choosing the concentration of Carbopol® 940 in emulgels, we primarily focused on the literature data. In a significant part of the published works, Carbopol® 940 is used as a thickener at a concentration of 0.5% wt [10–12]. The emulgel obtained by us, containing 0.5% wt Carbopol® 940, was homogeneous, fairly viscous, and was stable for a long time (more than a month). The consistence of the resulting emulgel was suitable in terms of rheological characteristics for in vivo experiments.
This information is given in revised manuscript, page 4.
- FewDLSdataareavailableinthemanuscript. Meanwhiletheyareimportantnotonlyforcharacterizingtheparticlesizes, butalsoformonitoringtheirstability. IrecommendtotransferthefigureortablewithdimensionalcharacteristicsfromSIsectiontothemainpartofthemanuscript, andgivedetailedinformationonthismethodintheexperimentalpart.
Author response:
Thank you for your recommendation. We moved the table with the characteristics of microemulsions sizes from Supplementary materials to the discussion of the results. We also improved experimental section regarding particle size determination. Changes in the text of the Manuscript are marked by blue type.
- Forconvenience, Irecommendtomovethetable, whichshowsthecompositionsofthemicroemulsionsandemulgels, aswellastheirdesignations, fromtheexperimentalparttothediscussionoftheresults.
Author response:
Thank you for your recommendation. We moved the table, which shows the compositions of the microemulsions and emulgels from the experimental part to the discussion of the results.
Minor:
- “As a result of comparative experiments, in which the size and degree of polydispersity of particles in the system, its stability and solubilization capacity with respect to abietic acid, as well as the ease of its conversion into emulgels, were evaluated, twomain microemulsions (ME 1 and ME 3), composition for which are given in Table 1.”
It should be table 2, please revise.
Author response:
Thank you very much. We fixed table numbering according to their order in the manuscript. In the revised manuscript, the correct Table number is indicated.
- It follows from the DLS data that the introduction of a cationic surfactant into the initial microemulsion leads to some decrease in the particle size, but the addition of abietic acid practically does not change their hydrodynamic diameter (Table S1, Figure S1,S2).
It should be Figures S1,S2.
Author response:
Thank you very much. We revised the text.
- In the table S1 extinction coefficient is equal to 12700, but in the text the value of 13000 is given? Bring these in line.
Author response:
Since abietic acid is insoluble in water, experiments on its release were carried out in a PBS:ethanol (1:1) solution. In this mixture, extinction coefficient was found to be 13000 l⋅mol−1⋅cm−1. We added this value to table S1 (it was originally S2, but since the table with size characteristics was transferred to the text of the article, the numbering has changed). We have made the appropriate correction in the text.
- Did the authors measure the cumulative release? Add details to Materials and Methods if applicable.
Author response:
Thank you, yes, we did evaluate cumulative release using dialysis technique. It was not clearly stated in the methods section, so we added a bit of text for easier reading in the methods section 3.6.
We thank the Reviewer for their helpful revision work. We have addressed all the issues raised by them.
References
- Wender, P.A.; Rothbard, J.B.; Jessop, T.C.; Kreider, E.L.; Wylie, B.L. Oligocarbamate Molecular Transporters: Design, Synthesis, and Biological Evaluation of a New Class of Transporters for Drug Delivery. J. Am. Chem. Soc. 2002, 124, 13382–13383, doi:10.1021/ja0275109.
- Zhou, H.; Yang, J.; Du, Y.; Fu, S.; Song, C.; Zhi, D.; Zhao, Y.; Chen, H.; Zhang, S.; Zhang, S. Novel Carbamate-Linked Quaternary Ammonium Lipids Containing Unsaturated Hydrophobic Chains for Gene Delivery. Bioorganic & Medicinal Chemistry 2018, 26, 3535–3540, doi:10.1016/j.bmc.2018.05.029.
- Liu, D.; Zhang, H.; Wang, Y.; Liu, W.; Yin, G.; Wang, D.; Li, J.; Shi, T.; Wang, Z. Design, Synthesis, and Biological Evaluation of Carbamate Derivatives of N-Salicyloyl Tryptamine as Multifunctional Agents for the Treatment of Alzheimer’s Disease. European Journal of Medicinal Chemistry 2022, 229, 114044, doi:10.1016/j.ejmech.2021.114044.
- Kushnazarova, R.A.; Mirgorodskaya, A.B.; Lukashenko, S.S.; Voloshina, A.D.; Sapunova, A.S.; Nizameev, I.R.; Kadirov, M.K.; Zakharova, L.Ya. Novel Cationic Surfactants with Cleavable Carbamate Fragment: Tunable Morphological Behavior, Solubilization of Hydrophobic Drugs and Cellular Uptake Study. Journal of Molecular Liquids 2020, 318, 113894, doi:10.1016/j.molliq.2020.113894.
- Mirgorodskaya, A.B.; Koroleva, M.Y.; Kushnazarova, R.A.; Mishchenko, E.V.; Petrov, K.A.; Lenina, O.A.; Vyshtakalyuk, A.B.; Voloshina, A.D.; Zakharova, L.Y. Microemulsions and Nanoemulsions Modified with Cationic Surfactants for Improving the Solubility and Therapeutic Efficacy of Loaded Drug Indomethacin. Nanotechnology 2022, 33, 155103, doi:10.1088/1361-6528/ac467d.
- Ghosh, A.K.; Brindisi, M. Organic Carbamates in Drug Design and Medicinal Chemistry. J. Med. Chem. 2015, 58, 2895–2940, doi:10.1021/jm501371s.
- Matošević, A.; Radman Kastelic, A.; Mikelić, A.; Zandona, A.; Katalinić, M.; Primožič, I.; Bosak, A.; Hrenar, T. Quinuclidine-Based Carbamates as Potential CNS Active Compounds. Pharmaceutics 2021, 13, 420, doi:10.3390/pharmaceutics13030420.
- Safitri, F.I.; Nawangsari, D.; Febrina, D. Overview: Application of Carbopol 940 in Gel: In Proceedings of the Proceedings of the International Conference on Health and Medical Sciences (AHMS 2020); Atlantis Press: Yogyakarta, Indonesia, 2021.
- Ajazuddin; Alexander, A.; Khichariya, A.; Gupta, S.; Patel, R.J.; Giri, T.K.; Tripathi, D.K. Recent Expansions in an Emergent Novel Drug Delivery Technology: Emulgel. Journal of Controlled Release 2013, 171, 122–132, doi:10.1016/j.jconrel.2013.06.030.
- Teaima, M.H.; Badawi, N.M.; Attia, D.A.; El-Nabarawi, M.A.; Elmazar, M.M.; Mousa, S.A. Efficacy of Pomegranate Extract Loaded Solid Lipid Nanoparticles Transdermal Emulgel against Ehrlich Ascites Carcinoma. Nanomedicine: Nanotechnology, Biology and Medicine 2022, 39, 102466, doi:10.1016/j.nano.2021.102466.
- Algahtani, M.S.; Ahmad, M.Z.; Shaikh, I.A.; Abdel-Wahab, B.A.; Nourein, I.H.; Ahmad, J. Thymoquinone Loaded Topical Nanoemulgel for Wound Healing: Formulation Design and In-Vivo Evaluation. Molecules 2021, 26, 3863, doi:10.3390/molecules26133863.
- Algahtani, M.S.; Ahmad, M.Z.; Ahmad, J. Nanoemulgel for Improved Topical Delivery of Retinyl Palmitate: Formulation Design and Stability Evaluation. Nanomaterials 2020, 10, 848, doi:10.3390/nano10050848.
- Düwel, S.; Hundshammer, C.; Gersch, M.; Feuerecker, B.; Steiger, K.; Buck, A.; Walch, A.; Haase, A.; Glaser, S.J.; Schwaiger, M.; Schilling F. Imaging of pH in vivo Using Hyperpolarized 13C-labelled Zymonic Acid. Nat. Commun. 2017, 8, 15126; DOI:10.1038/ncomms15126.
- Gupta, R.K. Polymer and Composite Rheology, 2nd ed.; Marcel Dekker, Inc.: New York, NY, USA, 2000; ISBN 0-8247-9922-4.

Reviewer 2 Report
The paper entitled “Supramolecular tools to improve wound healing and antioxidant properties of abietic acid: biocompatible microemulsions and emulgels” is interested and contains some interesting information about abietic acid as antioxidant ingredient in pharmaceutical applications. Also, the paper showed development of drug delivery systems based on microemulsions and emulgels. My opinion is that the paper can be published in Molecules after revision. I suggest to authors to explain, in Material and methods, methods for oscillatory rheological measurements presented in Figure 1. It is necessary to write how they did determine linear viscoelastic region as well as method for frequency sweep test. It would be necessary to present frequency sweep test results for other emulges as well as microemulsions and make some conslusions. Also, it would be important to enter results for G”/G’ ratio in order to conclude about three-dimensional structure. The dynamic viscosity is wrong term for apparent viscosity (Figure S4). The discussion about droplets size and distribution of microemulsions as well as z-potential is very poor. The difference in disperse characteristics and z-potential needs to be related to presence of different ingredients.
Author Response
Response of authors to reviewer comments
Manuscript ID: molecules-1918422
Title: Supramolecular tools to improve wound healing and antioxidant properties of abietic acid: biocompatible microemulsions and emulgels
Journal: Molecules
Corresponding Author: Dr. Kushnazarova
Authors: Alla Mirgorodskaya, Rushana Kushnazarova*, Rais Pavlov, Farida Valeeva, Oxana Lenina, Ksenia Bushmeleva, Dmitry Kuryashov, Alexandra Vyshtakalyuk, Gulnara Gaynanova, Konstantin Petrov, Lucia Zakharova
Reviewer 2
The paper entitled “Supramolecular tools to improve wound healing and antioxidant properties of abietic acid: biocompatible microemulsions and emulgels” is interested and contains some interesting information about abietic acid as antioxidant ingredient in pharmaceutical applications. Also, the paper showed development of drug delivery systems based on microemulsions and emulgels. My opinion is that the paper can be published in Molecules after revision.
- I suggest to authors to explain, in Material and methods, methods for oscillatory rheological measurements presented in Figure 1. It is necessary to write how they did determine linear viscoelastic region as well as method for frequency sweep test.
Author response:
Thank you for your recommendation. We have made appropriate clarifications and additions in Material and methods.
- 2. It would be necessary to present frequency sweep test results for other emulgels as well as microemulsions and make some conslusions. Also, it would be important to enter results for G”/G’ ratio in order to conclude about three-dimensional structure.
Author response:
Thank you very much. We conducted additional studies and compared the rheological behavior of microemulsions and emulgels created on their basis. Figure 1a illustrates the evolution of SAOS (small amplitude oscillatory shear) functions with frequency, at 25 °C, inside the linear viscoelastic range for ME4 and EG4.
|
|
|
|
Figure 1. Frequency dependence of the storage, G’, and loss, G” (a), moduli and the loss tangent for ME 4 and EG 4 (b). |
|
As can be observed, different responses were obtained for the systems under study. In the case of the microemulsion (ME4), the loss modulus (G″) are significantly higher than those for the storage modulus (G′), and the formulations are not really resulting in gels but liquid-like dispersions. A tendency to reach a crossover point between these moduli was observed at high frequencies. This corresponds to the dynamic characteristics of a fluid without entanglements. On the other hand, for EG4, G′ becomes higher than G″ and is practically unchanged in the whole frequency range. This indicates the formation of elastic gels. As known [1], the gel strength of biopolymer dispersed systems, from dilute solutions to crosslinked gels, can be quantified from SAOS measurements as a function of the G’ and G″ frequency dependence. In this case, characteristic parameters are the slopes of G’ and G” versus frequency plots, and the relative values of both viscoelastic functions, i.e., the relative elasticity, expressed in terms of the loss tangent (tan δ = G″/G’). Regarding the evolution of the loss tangent with frequency (Figure 1b), the EG4 shows the lower values of the loss tangent indicating a higher relative elasticity due to the high-level physical entanglements. On the other hand, ME4 displays the highest values of the loss tangent, higher than 1 in the whole frequency range indicating an essentially viscous behavior characteristic of polymer fluids without significant formation of physical entanglements [2]. We have made appropriate additions to the manuscript.
- 3. The dynamic viscosity is wrong term for apparent viscosity (Figure S4).
Author response:
Thanks, the term has been corrected (Figure S6).
- The discussion about droplets size and distribution of microemulsions as well as z-potential is very poor. The difference in disperse characteristics and z-potential needs to be related to presence of different ingredients
Author response:
As requested by the Reviewer we have expanded the part of the article regarding droplets size and distribution of microemulsions as well as zeta potential. In Table 2, we have presented data characterizing the size and degree of polydispersity (PdI) of particles. We supported the discussion of the values of the zeta potential with figures (Figures S3, S4). In addition, we added some details of the DLS measurements in the experimental part.
We thank the Reviewer for their helpful revision work. We have addressed all the issues raised by them.
References
- Düwel, S.; Hundshammer, C.; Gersch, M.; Feuerecker, B.; Steiger, K.; Buck, A.; Walch, A.; Haase, A.; Glaser, S.J.; Schwaiger, M.; Schilling F. Imaging of pH in vivo Using Hyperpolarized 13C-labelled Zymonic Acid. Nat. Commun. 2017, 8, 15126; DOI:10.1038/ncomms15126.
- Gupta, R.K. Polymer and Composite Rheology, 2nd ed.; Marcel Dekker, Inc.: New York, NY, USA, 2000; ISBN 0-8247-9922-4.

Round 2
Reviewer 2 Report
In my opinion authors made an effort to improve the previous version of Manuscript. So, I will recommend to Editor to accept this paper.